# High Clinical Value of Liquid Biopsy to Detect Circulating Tumor Cells and Tumor Exosomes in Pancreatic Ductal Adenocarcinoma Patients Eligible for Up-Front Surgery

**DOI:** 10.3390/cancers11111656

**Published:** 2019-10-26

**Authors:** Etienne Buscail, Catherine Alix-Panabières, Pascaline Quincy, Thomas Cauvin, Alexandre Chauvet, Olivier Degrandi, Charline Caumont, Séverine Verdon, Isabelle Lamrissi, Isabelle Moranvillier, Camille Buscail, Marion Marty, Christophe Laurent, Véronique Vendrely, François Moreau-Gaudry, Aurélie Bedel, Sandrine Dabernat, Laurence Chiche

**Affiliations:** 1U1035 Institut National de la Santé et de la Recherche Médicale (INSERM), 33000 Bordeaux, France; ebuscail@me.com (E.B.); pascaquincy@gmail.com (P.Q.); cauvin.lapeyre@gmail.com (T.C.); alexandre.chauvet@yahoo.fr (A.C.); olivier.degrandi@chu-bordeaux.fr (O.D.); isabelle.lamrissi-garcia@u-bordeaux.fr (I.L.); isabelle.moranvillier@u-bordeaux.fr (I.M.); christophe.laurent@chu-bordeaux.fr (C.L.); veronique.vendrely@chu-bordeaux.fr (V.V.); francois.moreau-gaudry@u-bordeaux.fr (F.M.-G.); aurelie.bedel@u-bordeaux.fr (A.B.); laurence.chiche@chu-bordeaux.fr (L.C.); 2Centre Hospitalier Universitaire (CHU) de Bordeaux, 33000 Bordeaux, France; charline.caumont@chu-bordeaux.fr (C.C.); severine.verdon@chu-bordeaux.fr (S.V.); marion.marty@chu-bordeaux.fr (M.M.); 3Université de Bordeaux, 33076 Bordeaux, France; 4Laboratory of Rare Human Circulating Cells, University Medical Centre of Montpellier, EA2415 Montpellier, France; c-panabieres@chu-montpellier.fr; 5Nutritional Epidemiology Research Team (EREN), Paris 13 University, U1153 INSERM, U1125 Institut national de la recherche agronomique (INRA), Conservatoire national des arts et métiers (CNAM), Paris Cité Epidemiology and Statistics Research Center (CRESS), 93017 Bobigny, France; camille.debrauer@gmail.com

**Keywords:** pancreatic cancer, liquid biopsy, exosomes, circulating tumor cells

## Abstract

Purpose: Expediting the diagnosis of pancreatic ductal adenocarcinoma (PDAC) would benefit care management, especially for the start of treatments requiring histological evidence. This study evaluated the combined diagnostic performance of circulating biomarkers obtained by peripheral and portal blood liquid biopsy in patients with resectable PDAC. Experimental design: Liquid biopsies were performed in a prospective translational clinical trial (PANC-CTC #NCT03032913) including 22 patients with resectable PDAC and 28 noncancer controls from February to November 2017. Circulating tumor cells (CTCs) were detected using the CellSearch^®^ method or after RosetteSep^®^ enrichment combined with CRISPR/Cas9-improved *KRAS* mutant alleles quantification by droplet digital PCR. CD63 bead-coupled Glypican-1 (GPC1)-positive exosomes were quantified by flow cytometry. Results: Liquid biopsies were positive in 7/22 (32%), 13/22 (59%), and 14/22 (64%) patients with CellSearch^®^ or RosetteSep^®^-based CTC detection or GPC1-positive exosomes, respectively, in peripheral and/or portal blood. Liquid biopsy performance was improved in portal blood only with CellSearch^®^, reaching 45% of PDAC identification (5/11) versus 10% (2/22) in peripheral blood. Importantly, combining CTC and GPC1-positive-exosome detection displayed 100% of sensitivity and 80% of specificity, with a negative predictive value of 100%. High levels of GPC1^+^-exosomes and/or CTC presence were significantly correlated with progression-free survival and with overall survival when CTC clusters were found. Conclusion: This study is the first to evaluate combined CTC and exosome detection to diagnose resectable pancreatic cancers. Liquid biopsy combining several biomarkers could provide a rapid, reliable, noninvasive decision-making tool in early, potentially curable pancreatic cancer. Moreover, the prognostic value could select patients eligible for neoadjuvant treatment before surgery. This exploratory study deserves further validation.

## 1. Introduction

Whereas overall survival of pancreatic ductal adenocarcinoma (PDAC) is less than 10%, survival can reach around 20% when surgery is possible, giving the best chance to the patients [1]. The diagnosis of PDAC can be challenging, especially for patients eligible for up-front surgery. Imaging is the first diagnostic tool used to decide resectability in patients who are strongly suspected to have pancreatic cancers [2]. Patients with small lesions, a hypertrophied pancreatic head, isoattenuating tumors, and focal fatty infiltration of the parenchyma might necessitate further investigations. Echo-endoscopy ultrasound guided-fine-needle aspirations (EUS-FNA) are strongly recommended, as they represent the sole tool able to diagnose the malignity of the lesion [3]. However, conventional tissue biopsies show heterogeneous diagnostic performance because of the intrinsic nature of the tumors with low cellularity associated with high stromal content. Moreover, they are operator-dependent. These difficulties lead to noninformative analysis of the tumor and even to false-negative diagnosis, with a negative predictive value ranking between 33% and 85%. Overall, this test may be inconclusive or doubtful in up to 20% of cases [4]. The alternative circulating biomarkers, such as the serum protein markers CEA(carcinoembryonic antigen*)* and CA19.9, are used to monitor early recurrences, but their low sensitivity and specificity prevent any use as screening or diagnostic tools [5].

Primary tumors release in the blood and other bodily fluids complex tumor-derived elements, such as circulating tumor cells (CTCs) and exosomes. When identified, these circulating biomarkers could be considered as a proof of the presence of the tumor for various cancers [6], including PDAC [7]. *Liquid biopsy* might represent a noninvasive, safe, and fast companion test to tissue biopsy [8]. CTC detection has been carried out with diverse nonequivalent approaches that could be complementary in improving the CTC detection rate. In particular, the most popular method is the CellSearch^®^ system because it has been cleared by the United States FDA(food and drug administration) to monitor metastatic prostate, breast, and colorectal cancers [9,10,11]. However, CellSearch^®^ may not detect CTCs that have undergone the epithelial-to-mesenchymal transition (EMT). Thus, alternative methods have been developed. The density gradient centrifugation with OncoQuick^®^ resulted in higher relative tumor-cell enrichment than the Ficoll density gradient centrifugation [12] and provided a good detection rate of EpCAM-negative breast cancer CTCs [13]. Another EpCAM unbiased approach is to negatively enrich blood samples with CTCs by using immune cocktails to withdraw the blood mononuclear cells [14,15]. CTC-enrichment methods have been combined with molecular identification such as the detection of mutant *KRAS*, present in >92% of PDACs [16].

The tumor-released exosomes raised high interest because they carry the physiopathological signature of the emitting cells, not only via molecules present in their membranes, but also via components they carry [17]. In particular, PDAC exosomes carry the membrane heparan sulfate proteoglycan Glypican-1 (GPC1) that detected 100% of patients with PDAC and distinguished patients with precancerous pancreatic lesions from those with benign diseases [18]. However, recent studies [19,20,21], including ours [22], found more moderate power of GPC1-based exosome quantification for PDAC diagnostics.

In general, studies with high diagnostic values of *liquid biopsy* include a majority of patients with advanced disease, with potentially more circulating tumor elements. In this study, we aimed to assess whether combining methods for CTC detection and PDAC exosomes was efficient for PDAC diagnostics and carried prognostic value in a homogeneous group of patients with an early stage disease, all eligible for up-front surgery. In addition, portal blood was previously found to contain numerous CTCs as compared to peripheral blood in patients with advanced disease [23,24], and even in patients with resectable tumors [25,26,27]. Thus, to increase chances of detecting CTCs and/or exosomes, we analyzed peripheral and portal blood samples.

## 2. Materials and Methods

### 2.1. Study Design

We enrolled patients eligible for pancreatic surgery with suspicion of pancreatic cancer without metastasis or suspicion of IPMN (intraductal papillary and mucinous neoplasm) with worrisome features. Diagnostics were performed by CT-scan (computerized tomography scanner) and/or MRI (magnetic resonance imaging). Patients were enrolled at the department of hepatobiliary surgery of Bordeaux university hospital between February and November 2017. All patients underwent standardized staging, including CT-scan, MRI (in case of doubt on liver metastasis), and CA19-9 as well as an evaluation by a multidisciplinary board. Exclusion criteria were borderline or locally advance diseases with an indication of neo-adjuvant therapy [2], metastatic disease, or history of other malignancies. The control group included patients who underwent a surgical procedure in our department for non-neoplastic pathology and without a history of solid cancer or hematologic malignancy. This prospective study was conducted according to the Declaration of Helsinki, the French rules (Law for Bioethics November 2016, article L.5311-1, code de la santé publique), and the recommendations of CNIL (Comité National Informatique et Liberté), and was approved by the Institutional Review Board, Comité de protection des Personnes Sud-Ouest et Outremer III. The biological collection was declared to and approved by the French Ministry of Research under the number 2016-A00431-50, and the database was registered in Clinical Trials under the number NCT03032913. Informed written consent and information were obtained from patients before surgery. Patient follow-up was done until 1 December, 2018. A scan was performed 3 and 6 months after surgery with CA19-9 dosage each time. Postoperative data were also collected.

### 2.2. Surgical Procedure, Blood Sampling, and Tumor Staging

After laparotomy, we inspected and palpated the liver and peritoneal cavity to identify metastasis. Frozen sections were performed for suspicious lesions, and resection was abandoned if intra-operative specimens were positive for metastatic adenocarcinoma. Pancreaticoduodenectomy (Whipple procedure) was started with isolation and division of the common bile duct, and then the portal vein was exposed. For left pancreatectomy, a retro-isthmic puncture of the portal vein was performed. Two samples of 7.5 mL of blood were collected from the portal vein in BD vacutainer collection tubes without additives (SST tubes, Becton Dickinson, Le Pont de Claix, France) before any manipulation of the tumor. For the patient group, additional samples of 7.5 mL were collected during surgery from the median cephalic vein in BD vacutainer collection tubes without additives. For all individuals, a sample of 7.5 mL in a BD vacutainer collection tube containing EDTA for the RosetteSep^TM^-based CTC enrichment and, for the PDAC group only, a 7.5 mL sample in a CellSave tube (Menarini Silicon Biosystems Inc., Castel Maggiore, Italy) for the CellSearch^®^ procedure were also collected. Tubes were transferred quickly in the laboratory and were centrifuged at 2000 g for 15 min to collect sera to determine CA19-9 concentration (Architect automated instrument, Abbott, Chicago IL, USA). Sera were frozen at −80 °C until they were further processed for exosome quantification. Tumor staging was performed according to the TNM AJCC2017, 8th version, and histological analyses were performed by a single specialized pathologist [28]. Progression-free survival (PFS) and overall survival (OS) were defined by the time from surgery to progression based on CT staging or to death, respectively.

### 2.3. CTC Identification

First, the RosetteSep^TM^ (Stemcell technologies, Grenoble, France) and OncoQuick^®^ (Greiner Bio One SAS, Les Ullis, France) capacities to recover tumor cells from total blood samples were tested using CAPAN-2 cell line spiking experiments. Total blood from healthy volunteers was obtained from the Etablissement Français du sang (EFS, Pr. Jeanne, convention 16PLER023). CAPAN-2 cells were first transduced with the pSIN-EF1aL-eGFP-IRES-Puro lentivector (Vect’UB, Bordeaux, France), and the subpopulation of green fluorescent cells was sorted on a FACSARIA II (BD Biosciences, Le Pont de Claix, France). A known number of fluorescent cells (15–42 for RosetteSep^TM^ and 11–44 for OncoQuick^®^) were spiked in 7.5 mL of blood from healthy donors. CTC enrichment was carried out according to the manufacturers’ protocols. Cell pellets were recovered in a minimal volume of cell medium in 60-well plates (Thermofisher, Courtaboeuf, France) in order to recover all the cells in a single well. Fluorescent cells were counted under an inverted Nikon Microscope (Eclipse Ti Nikon, Champigny sur Marne, France). Pictures were taken with the NIS-Elements Nikon software (Minato, Tokyo, Japan), connected to a video camera.

Patient and control total blood samples were enriched in the same way. Cell pellets were further analyzed for the presence of *KRAS* mutations by droplet digital PCR (ddPCR), with the *KRAS* G12/G13 Screening Kit (Biorad, Marne la Coquette, France) after total DNA extraction with the RSC ccfDNA plasma kit, Maxwell (Promega, Charbonnières-les-Bains, France).

The CellSearch^®^ semi-automated platform with the Circulating Epithelial Cell Kit and the CELLTRACKS ANALYZER ll System were used for CTC detection (Menarini Silicon Biosystems Inc., Castel Maggiore, Italy). Blood sample CellSave tubes were kept at room temperature until processing, which was completed within 36 h. After automated EpCAM-based immunomagnetic sorting, all objects presented on the CellSearch^®^ screen were analyzed by a certified technician. All cells meeting the CellSearch^®^ analysis standards for CTCs (DAPI^+^, CK^+^, EpCAM^+^, and CD45^−^ with a cellular shape and visible nucleus) were counted, and the final diagnostic approval was done by a single experienced biologist. The presence of 1 CTC/7.5 mL was considered positive as previously described [23,26,29].

### 2.4. CRISPR/Cas9-Driven Cut of KRAS Wild-Type (WT) Allele

Ribonucleoproteic complexes (RNPs) containing 104 pmol of Cas9 and 120 pmol of the WT *KRAS* specific guide RNA (5′GGAAACTTGTGGTAGTTGGAGC GUUUUAGAGCUAGAAAUAGCAAGUUAAAAUAAGGCUAGUCCGUUAUCAACUUGAAAAAGUGGCACCGAGUCGGUGCUUUUU 3′, [30]) in 5 µL, were prepared at the final concentration of 1 µg/µL of Cas9. DNAs extracted from RosetteSep^TM^-enriched CTC pellets (see above) were treated with 1 µg of RNPs for 18 h at 37 °C. The cut DNA (2 µL) was then amplified by conventional PCR (GoTaq^®^, Promega, Charbonnières-les-Bains, France, 45 cycles at 50 °C, forward primer 5′-GGTGAGTTTGTA TTA AAA GGT ACT GG-3′ and reverse primer 5′-TCCTGCACCAGTAATATGCA-3′), followed by ddPCR of 50 ng of PCR product, with the *KRAS* G12/G13 Screening Kit, according to the manufacturer’s instructions.

### 2.5. Exosome Analysis

This procedure has been recently published by our group [22]. In brief, sera were enriched in extracellular vesicles (EVs) using the Total Exosome Isolation kit (Thermofisher), according to the manufacturer’s instructions. The Exosome-Human CD63 Isolation/Detection Reagent (Thermofisher) was used to pull down sera exosomes. They were stained with anti-GPC1 primary antibody (PIPA528055, Thermofisher) and revealed with Alexa Fluor 647 donkey anti rabbit IgG (Biolegend, San Diego, CA, USA) on a BD FACS CANTO II apparatus (BD Biosciences). Percentages of GPC1-positive beads were determined with BD FACS Diva software (BD Biosciences).

### 2.6. Statistics

Characteristics of both groups were compared using Fisher’s exact test or Wilcoxon–Mann–Whitney’s test according to the type of data (qualitative or quantitative, respectively). The OS and PFS were first determined by the Kaplan–Meier method. A log-rank test was then used to assess the associations between various covariates and OS. Student’s *t* test was used to compare tumor cell recovery rates in cell spiking experiments. All statistical analyses were performed using the GraphPad-Instat and GraphPad-Prism 8.0 software programs (GraphPad Software Inc. San Diego, CA, USA). A *p* value < 0.05 was considered significant.

## 3. Results

### 3.1. Cohort Characteristics

Seventy-two patients underwent surgery for PDAC from February to November 2017. Upfront surgery was performed for 32 patients for presumed PDAC without neo-adjuvant therapy (Figure 1). Among them, two metastatic patients were excluded. Eight patients were excluded from the cancer group and switched to the control group after definitive pathology analysis, because of noninvasive intraductal papillary mucinous neoplasm (IPMN) diagnosis. Thus, the control group included 28 control patients without neoplasia and without a history of cancer. They consisted of two chronic pancreatitis patients with surgery for symptomatic reasons operated on in our surgical unit, eleven cholecystectomies, three bariatric procedures, two hernia surgeries, two functional pelvic floor surgeries, and the eight IPMNs mentioned above. Demographic and clinico-pathological characteristics were similar between groups except for age, since PDAC patients were significantly older than individuals in the control group (Appendix A). Patients underwent 20 Whipple procedures and two left pancreatectomies. The IPMN control group consisted of seven Whipple procedures and one left pancreatectomy. Mean tumor size was 31 mm. Tumor stages were 22.5% stage I, 50% stage IIb, and 27.5% stage III. Positive lymph nodes were found for 77.5% of the PDAC patients (Table 1).

### 3.2. Cell Spiking Experiments

CTC counting with CellSearch^®^ is limited to the identification of EpCAM^+^ pancreatic tumor cells. To increase our chances of detecting EpCAM negative cells, we tested two CTC enrichment methods followed by *KRAS* mutant DNA detection by ddPCR. First, known numbers of GFP^+^–CAPAN-2 cells were spiked into 7.5 mL of healthy donor total blood samples. Percentages of spiked cell recovery were determined by fluorescent cell counting under a microscope after spiked blood samples were processed with RosetteSep^TM^ or OncoQuick^®^ to obtain blood cell pellets enriched with CTCs (Appendix A). All experiments allowed for the isolation of at least one tumor cell, but percentages of recovery were higher in OncoQuick^®^ as compared to RosetteSep^TM^ (Appendix A, 67.5% ± 3.5%, *n* = 59 and 50.7% ± 3.5%, *n* = 65, respectively, *p* < 0.001). However, cell enrichment was 10 times lower, as determined by total cell count after recovery (not shown), leading to high levels of contamination mainly with PBMCs (Peripheral blood mononuclear cells) (Appendix A). Molecular detection of mutant *KRAS* alleles by ddPCR after RosetteSep^TM^ enrichment was 3- to 4-fold more sensitive than after OncoQuick^®^ (Appendix A).

Thus, OncoQuick^®^ was superior to RosetteSep^TM^ in recovering tumor cells, but RosetteSep^TM^ was more sensitive in detecting tumor DNA. Therefore, we used RosetteSep^TM^ to enrich PDAC patient plasmas in CTCs. Of note, all primary tumors displayed mutant *KRAS* alleles by ddPCR (Table 1).

### 3.3. Diagnostic Values of CTC or GPC1^+^-Exosome Detection in Peripheral and Portal Blood

Each patient was subjected to CTC detection by two independent methods. Interestingly, CellSearch^®^ identified five out of 11 patients (46%) for whom we had portal blood samples and only 2/22 (9%) when peripheral blood was considered. Thus, CTC detection with CellSearch^®^ showed an expected low sensitivity of 32% and a very strong specificity of 100% (the IPMN group was considered the control group, as the CellSearch^®^ technique was not performed on the noncancer group) (Table 2, Figure 2A,B). Of note, patients (#49 and #50, 2/11) displayed cell clusters in portal blood (Figure 3C).

Direct *KRAS* mutant detection by ddPCR after RosetteSep^TM^ CTC enrichment displayed a low PDAC identification rate (2/22, 9%) in peripheral blood. All controls were negative. However, when analyzing the raw ddPCR data, we found that 17/22 PDAC patients had MAFs (mutant allele frequencies) borderline to the detection threshold in at least one sample (portal or peripheral) (Figure 4A). Thus, we hypothesized that increasing the sensitivity of the PCR might better identify the PDAC patients, especially because we knew that PBMCs, bearing the wild-type (WT) *KRAS* allele, contaminated the CTC-enriched pellets. To address that, depleting the WT alleles with a CRISPR/Cas9-directed specific double-stranded cut was a good option to increase the chance of detecting mutant alleles [30]. All DNAs extracted after RosetteSep^TM^ enrichment (PDAC patients and controls) were analyzed again after Cas9-cut PCR. Out of the 17 samples, 11 became frankly positive (Figure 4A–D). Two previously negative PDAC samples were positive after Cas9 treatment. Thus, a total of 13/22 patients (59%, 11 in peripheral blood and 10 in portal blood, Figure 2A–F) were identified using Cas9-cut PCR/*KRAS* ddPCR. Four out of eight IPMN samples were found positive after Cas9-cut PCR treatment (3 in peripheral blood and 1 in portal blood, 50%), while two noncancer controls (10%) became positive (Figure 4C,D; Figure 2B,C). In fact, although gaining in sensitivity, specificity was affected by the positivity of four IPMNs (50%). In conclusion, RosetteSep^TM^-based CTC detection using Cas9-cut PCR (in portal and/or peripheral blood) was better than CA19-9 alone, with higher diagnostic accuracy (75% vs. 68%), explained by a better sensitivity of 59% versus 37%, respectively (Table 2). EUS-FNA carried a strong specificity, but a poor negative predictive value, as expected.

A total of 64% (14 out of 22) of PDAC patients had GPC1-positive exosomes in portal and/or peripheral blood (Figure 2A,D–F). Both sampling sites showed similar sensitivity and specificity (Table 2), and GPC1-positive exosomes in peripheral and/or portal blood displayed a diagnostic accuracy of 78%, similar to RosetteSep^TM^-based CTC detection (75%), and higher than CA19-9 (68%) or EUS-FNA (66%).

### 3.4. Diagnostic Value of Combined Diagnosis Methods

Overall, single liquid biopsy showed higher diagnostic performance than the routinely available tools (CA19-9 and EUS-FNA). Combining results from individual tools, in both sampling sites, proved to increase the number of detected patients, better than combining the traditional tools CA19-9 and EUS-FNA (Table 2: combined sensitivity). RosetteSep^TM^-based CTC detection and quantification of GPC1-positive exosomes displayed a very high sensitivity of 96%, with a high negative predictive value (96%), but false positives impacted the specificity (70%). Addition of CA19-9 and/or EUS-FNA did not improve the performances (Table 2). Noticeably, combining quantification of GPC1-positive exosomes, RosetteSep^TM^, and CellSearch^®^-based CTC detection identified all the PDAC patients (Figure 2F), showed a negative predictive value of 100%, and an overall diagnostic accuracy of 91% (Table 2). As four out of 20 noncancer controls were positive with either RosetteSep^TM^-based CTC detection or quantification of GPC1-positive exosomes, the specificity was 80% and the positive predictive value was 85% (Figure 2A–C; Table 2). Importantly, all the patients who were not diagnosed by EUS-FNA were identified by one or more liquid biopsies (Figure 2A).

### 3.5. Liquid Biopsy and Prognostic Performances

The PFS and median OS were 365 days (range 58–587) and 503 days (range 74–718), respectively. Individually, each biomarker was not prognostic. Interestingly, patients with >20% GPC1-positive exosomes in peripheral blood, which is 4 times the median value, and/or CellSearch^®^ positive clusters in portal blood had shortened OS (Figure 3A,C). Patients with GPC1-positive exosomes in peripheral blood and/or CellSearch^®^ positive CTC in peripheral blood had shortened PFS (Figure 3B). The tumor burden, in particular, tumor stage, node status, or tumor size, did not correlate with any individual liquid biopsy. Furthermore, there was no correlation between the tumor stage or the PFS or OS and the number of positive biomarkers.

## 4. Discussion

Our experimental design aimed to test the diagnostic performances of CTCs and GPC1-positive exosomes detected from peripheral and portal veins at identifying PDAC in a group of patients eligible for up-front surgery.

Taken individually, the CTC detection rates varied from 10% to 59%, in accordance with published results. In particular, previous CellSearch^®^ detection rates ranged from 11% to 48% in cohorts comprising at least 53% of patients with locally advanced or metastatic diseases (26,28). Including only patients with advanced diseases [31] did not increase rates of PDAC identification [32], suggesting that the CTC limit of detection is reached with the current protocol. A few studies analyzed patients with early stages of disease, similar to ours, and also found very low rates of PDAC identification. For example, 6.8% (2/37) with resectable tumors were detected in peripheral blood by CellSearch^®^ [33]. It was previously reported that CTCs are more numerous in portal blood, before they are filtered by the liver [34,35,36], for several cancers [25]. One hundred percent of patients with advanced or metastatic diseases had detectable CTCs in portal blood using either CellSearch^®^ or the similar ClearBridge^®^ systems [23,24]. When resectable patients were considered, numbers dropped to 49% and 58.5% [26,27]. This is in agreement with the identification of 5/11 (45%) patients by CellSearch^®^ in the portal blood in the present study. Thus, CellSearch^®^ has low capacity to detect PDAC patients in peripheral blood but is valuable when portal blood samples are considered, even in resectable patients considered to present early stage disease [37]. It might be interesting to test alternative methods of CTC detection, such as those recently published based on microfluidic platforms or based on cancer stem cell (CSC) detection [38].

Next, we tested an alternative CTC detection method based on the molecular identification of the *KRAS* mutant allele in CTC-enriched peripheral or portal blood samples. After RosetteSep^TM^-driven mononuclear blood cell depletion, we could identify only two samples with CTCs in the peripheral blood. However, after depleting DNA samples with the WT *KRAS* allele, about 50% of the samples displayed detectable mutant *KRAS*.

The last liquid biopsy-based PDAC identification tool we used was the quantification of GPC1-positive exosomes. We did not reproduce the sensitivity and specificity (100%) reported by Kalluri’s group, which were about 50% and 90%, respectively, regardless of the sampling site, as we recently discussed [22]. Others found detection rates close to ours. In particular, Yang et al. measured exosomal GPC1 levels by antibody-linked fluorescence intensity of ultracentrifuged plasma EVs, and found a sensitivity of 67% and a specificity of 82%. That group preferred a five-protein signature to increase their diagnostic accuracy [20]. Similarly, Lai et al. abandoned GPC1-based exosome quantification and reached 100% sensitivity and specificity with a combined approach of several microRNA signatures with GPC1 detection [19].

Various qualities of performances were reached with individual methods and we tested whether a combination of several methods could improve PDAC detection. Indeed, it is likely that early stage, resectable tumors release fewer circulating biomarkers, such as CTCs and exosomes. Combining all three different methods (quantification of GPC1-positive exosomes, RosetteSep^TM^, and CellSearch^®^-based CTC detection) identified 100% of the PDAC patients. The specificity was 80% because of a few false-positive controls as discussed below. Importantly, the negative predictive value was 100%, as compared to the poor negative predictive value of EUS-FNA (varying from 36% to 80% depending on the studies and confirmed here, with the challenging resectable tumors, 33% [4]). The high performance of combined biomarkers is interesting in the light of the current clinical practice evolution. If patients present locally advanced disease, it is crucial to establish as soon as possible the tumor’s resectability to avoid unneeded and even deleterious surgery for patients with undiagnosed metastatic stages [37]. Moreover, it is essential to increase the rate of complete (R0) resection with rapid and effective neoadjuvant therapy. Furthermore, neoadjuvant treatments, requiring histological and/or diagnostic evidence of tumor malignity, are now being considered, even when patients are directly resectable [39].

Diagnostic performance, especially specificity, has been reduced by false-positives in the control noncancer group. In particular, four out of eight IPMN samples were found positive with RosetteSep^TM^-based CTC enrichment and Cas9-cut ddPCR (50%) and two out of 20 noncancer controls (10%). This was not linked with age or smoking status of the patients. Previous studies using the ddPCR for identification of *KRAS* mutant alleles reported false-positive rates in exoDNA varying from 7.4% (4/54) to 20.7% and 25% (17/82 and 3/12) [40,41]. This might be partly explained by the fact that spontaneous somatic mutations are believed to occur in the normal population and healthy tissues [39,42,43] and by the high sensitivity of the PCR-based methods. The IPMN group showed interesting results, especially with RosetteSep^TM^-based CTC enrichment, because the positive patients were those with high grade dysplasia. Similarly, a previous report showed that CTC counts (by isolation by size of epithelial tumor cells) were higher in patients with high-grade dysplasia IPMN and were qualified as circulating epithelial cells (CECs) [44]. It would be very interesting to search for *KRAS* mutations in such CECs. The presence of false-positives is a limit of our study, but it does not invalidate the value of the combined approach. Indeed, the aim was to identify PDAC patients entering the care process with PDAC suspicion and not to screen the general population. In addition, due to the low lifetime risk of pancreatic cancer (around 1%), population-based screening of unselected individuals is not recommended for this tumor [45]. Two noncancer controls and one IPMN were false-positive for GPC1^+^-exosome quantification. Very interestingly, all were carriers of autoimmune pathologies (lupus or rheumatoid arthritis). It is possible that high levels of auto-antibodies interfere with the antibody-based test. Further investigations need to be carried out to test this hypothesis.

Besides us, only a few authors have investigated the value of detecting several biomarkers in a single *liquid biopsy* applied to PDAC. A combination of CTC detection (filtration-based method) and cell-free circulating tumor DNA (ctDNA) quantification, in association with CA19-9, identified 78% of PDACs, with a low negative predictive value (53%) [46]. Another recent report evaluated combined detection of ctDNA and exosomal DNA (exoDNA, *KRAS* mutant allele) on a prospective cohort of 168 patients with a majority of metastatic patients (60%). This approach identified 37.3% of metastatic PDACs and only 9.1% of patients with localized tumors [41], suggesting that the ctDNA approach is not suitable for resectable PDAC diagnosis. Indeed, we performed KRAS detection by ddPCR on our cohort and identified only the metastatic patients [22]. Instead, quantification of exoRNA (*KRAS* mutant) might be of high interest as it was recently reported for detecting mutant *EGFR* in lung cancer [47].

The detection rate of CTCs after the RosetteSep^TM^-enrichment step yielded similar efficiency to that observed by CellSearch^®^ in portal blood [26,27]. The same observation was made for exosome quantification, which was similar in both sample types (50% and 46%, respectively). So, unlike CellSearch^®^, these two detection methods were not impacted by the sampling site, suggesting that molecular-based detection methods suffer less from tumor-element dilution after liver filtration.

One of the strengths of our study is the homogeneity of the cohort reflecting the current clinical practice, especially the trickiest diagnostic situations. Few studies have produced homogeneous cohorts of patients with blood samples collected before any surgical and/or neo-adjuvant therapy. Instead, they included a majority of patients presenting advanced or metastatic diseases [19,21,29]. Our strength is also a limitation since the size of the cohort is small. However, we consider this study as a pilot study, worthy of further validation in bigger cohorts. Additionally, it would be of interest to test the combined methods for the diagnosis of all stages of disease and for longitudinal monitoring of on-treatment patients.

The presence of CTCs in the peripheral blood has been associated with a reduced PFS and OS in PDAC [6]. However, in our study, taken individually, none of the methods was associated with clinical outcomes or pathologic features. Interestingly, however, the combination of CTC detection by CellSearch^®^ and GPC1^+^-exosome quantification in peripheral blood was correlated to disease recurrence, and the high levels of exosomes combined with the presence of clusters was associated to OS. The presence of clusters has been found to correlate to a worse prognosis in other solid cancers [48,49].

Echo-endoscopy with puncture is the current gold standard for histologic proof and formal diagnostic of PDAC but has certain limitations. It carries variable negative predictive value and is largely operator-dependent for both the endoscopic ultrasound-guided-fine needle aspiration biopsy and the pathological analysis. It is invasive and while it carries a low morbidity rate, the risks are severe during the procedure, such as the possible induction of acute iatrogenic pancreatitis, which sometimes compromises surgical management [4]. With resectable disease, combined biomarkers in a single *liquid biopsy* could contribute to decision making, in particular, for triggering neo-adjuvant and adjuvant treatment.

## 5. Conclusions

In conclusion, our results suggest that combining biomarkers detection in *liquid biopsies* from peripheral and portal blood might represent a highly valuable diagnostic tool for patients with resectable PDAC. This study is exploratory and in need of further validation on a new cohort of patients with resectable tumors. Concomitant detection of several circulating tumor biomarkers, in other words, CTCs and exosomes, carried high diagnostic value and identified patients at risk of early disease relapses or fatal outcomes. This approach might greatly accelerate the diagnosis, which might in turn improve clinical outcomes and care experience. Adopting this approach, with a negative predictive value of 100%, might help decision making.

## Figures and Tables

**Figure 1 cancers-11-01656-f001:**
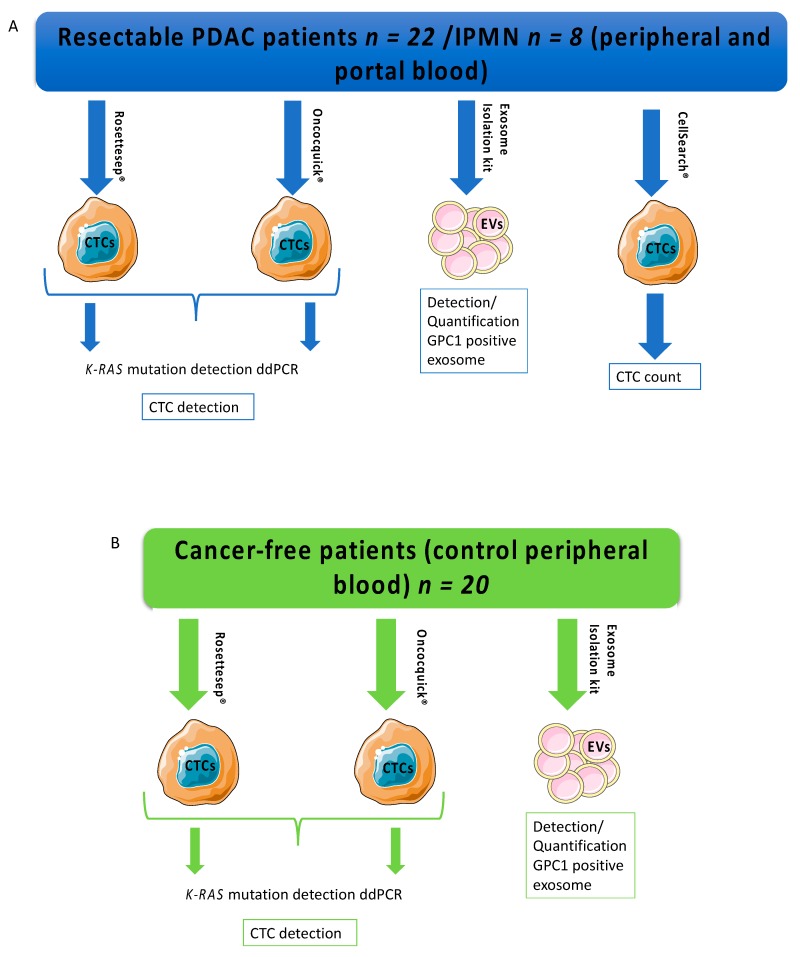
Study design, blood samples, and liquid biopsy methods. (**A**) Pancreatic ductal adenocarcinoma (PDAC) patients and patients with IPMN had both peripheral and portal samples for CTC-enrichment detection/count and quantification of GPC1-positive exosomes (blue rectangle and arrows). (**B**) Control group had peripheral samples for CTC-enrichment detection (RosetteSep^TM^) and quantification of GPC1-positive exosomes (green rectangle and arrows). Abbreviations: EVs: extracellular vesicles; CTC: circulating tumor cell; IPMN: intraductal papillary and mucinous neoplasm; GPC1: Glypican 1.

**Figure 2 cancers-11-01656-f002:**
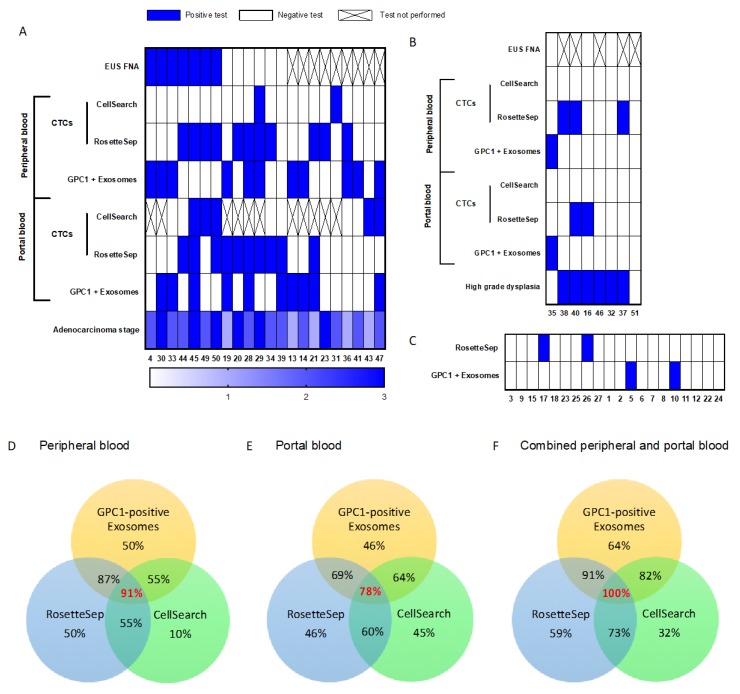
Heat maps of liquid biopsy results. (**A**) PDAC patients, (**B**) IPMN patients, and (**C**) noncancer control individuals. White rectangle: negative result, blue rectangle: positive result, crossed rectangle: not done. In the PDAC heat map, the bottom ladder indicates adenocarcinoma stage rankings from 1 to 3 according the stage of the disease (i.e., stage 1 light blue, stage 2 blue, stage 3 dark blue). In the IPMN heat map, the bottom ladder indicates dysplasia ranking from 0 (white box) for low grade dysplasia to 1 for high grade dysplasia (blue box). PDAC, pancreatic ductal adenocarcinoma; IPMN, intraductal papillary and mucinous neoplasm. (**D**–**F**) Venn diagrams recapitulating rates of CTC detection by CellSearch^®^ or RosetteSep^TM^-based enrichment and GPC1-positive-exosome quantification of (**D**) peripheral blood samples, (**E**) portal blood samples, (**F**) combined peripheral and portal blood samples.

**Figure 3 cancers-11-01656-f003:**
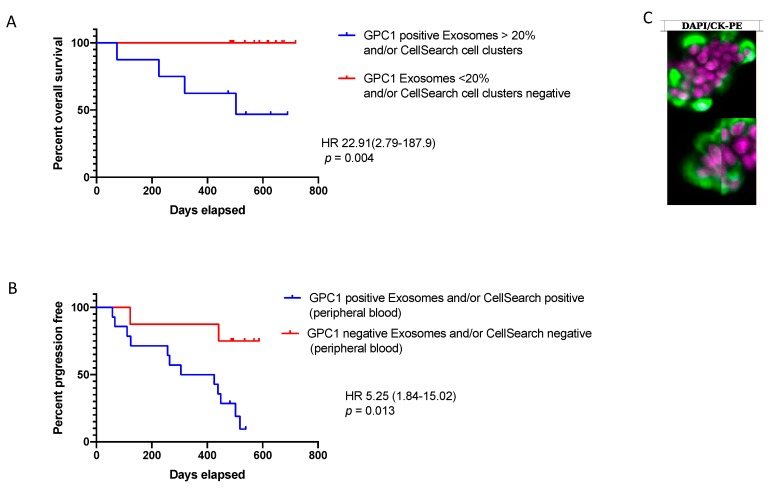
Analysis of GPC1-positive-exosome quantification and CellSearch^®^ positive CTC count and clusters according to clinical criteria. Kaplan–Meier curves, with *p* values (log Rank) for comparison between (**A**) overall survival (OS) for patients with >20% GPC1-positive exosomes (4 times the median value) and/or with CTC clusters and patient with <20% GPC1-positive exosomes and/or CellSearch^®^ without CTC clusters. (**B**) Progression-free survival (PFS) for patients with GPC1-positive exosomes and/or CellSearch^®^ positive and GPC1-negative exosomes and/or CellSearch^®^ negative in peripheral blood. (**C**) Immunofluorescent staining image of captured CTC clusters. Circulating tumor cell clusters captured from a portal vein sample using the CellSearch system. (CK, cytokeratin; PE, phycoerythrin; DAPI, 4′,6-diamidino-2-phenylindole; DAPI stain is purple and CK stain is green, (original magnification ×400)).

**Figure 4 cancers-11-01656-f004:**
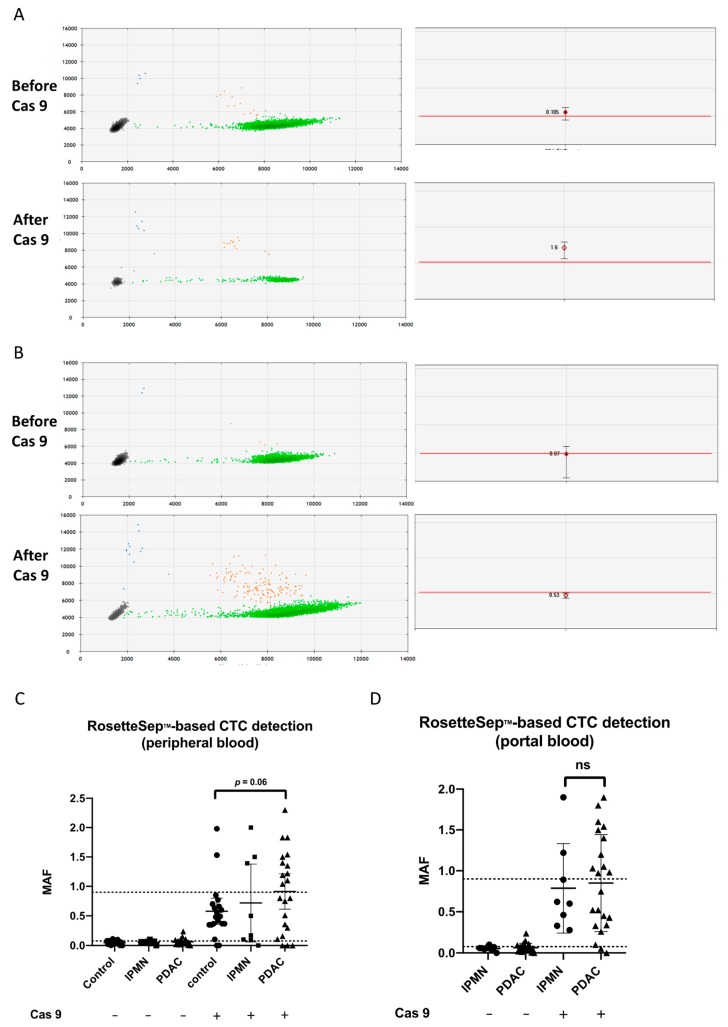
The ddPCR results for *KRAS* detection after CTC enrichment. (**A**,**B**) Individual droplet PCR fluorescence results are plotted as two-dimensional dot plots (left). Grey dots correspond to empty droplets. Green dots correspond to droplets containing wild-type (WT) copies of *KRAS*. Blue dots correspond to droplets containing one mutant *KRAS* allele. Orange dots correspond to droplets containing WT (*X*-axis of the left panels corresponding to the HEX, hexachlorofluorescein succinimidyl ester fluorophore) and mutant alleles (*Y*-axis of the left panels corresponding to the FAM, 6-carboxyfluoresceine fluorophore). On the right panels, MAFs are shown for individual results, with the maximum and the minimum values of triplicates; the red lines indicate the positivity threshold. Patient #36 (**A**) became positive and patient #39 (**B**) was negative for *KRAS* mutation before and after Cas9. (**C**,**D**) MAF of *KRAS* mutation by ddPCR after RosetteSep^TM^ CTC enrichment. Greater median MAF in CTC-enriched samples after CRISPR/Cas9 cut of the wild-type *KRAS* allele as compared to uncut DNA in (**C**) peripheral and (**D**) portal blood. Higher median MAFs in patients compared with the control group tended toward significance (*p* = 0.06 by Mann–Whitney test). MAF: mutant allele frequency.

**Table 1 cancers-11-01656-t001:** Details for pancreatic surgery and pathologic features (*n* = 30).

Variables	PDAC Group (*n* = 22) *n* (%)	IPMN (*n* = 8) *n* (%)
Procedures		
Whipple	20 (91)	7 (87)
Left pancreatectomy	2 (9)	1 (13)
Vascular reconstruction	4 (18)	0 (0)
Post-operative complications		
Dindo-Clavien III–IV	3 (13)	2 (25)
Dindo-Clavien V	0 (0)	0 (0)
Pathology: Macroscopic		
Tumor size (mm) mean (med; range)	31 (30; 11–49)	In situ carcinoma *n* = 0 (0) High grade dysplasia *n* = 6 (75) Low grade dysplasia *n* = 2 (25)
Tumor stage		
Stage 1a	1 (4.5)	
Stage 1b	4 (18)	
Stage 2b	11 (50)	
Stage 3	6 (27.5)	
Nodes status		
Positive	17 (77.5)	
Negative	5 (22.5)	
Glandular Differentiation		
Well	3 (13.5)	
Moderately	11 (50)	
Poorly	8 (36.5)	
*KRAS* status: all primary tumors were positive for *KRAS* mean mutant allele frequency (med; range)	26.15 (17.45; 0.35–77.6)	

Abbreviations: PDAC, Pancreatic ductal andenocarcinoma; med, median; IPMN, intraductal papillary and mucinous neoplasm. Note that Whipple surgery was performed for patients bearing tumors in the head of the pancreas, while left pancreatectomies were performed for patients with tumors in the tail of the pancreas.

**Table 2 cancers-11-01656-t002:** Diagnosis values of GPC1-positive exosomes, CTC detection by CellSearch^®^, and CTC quantification by RosetteSep^TM^, CA 19-9, and EUS-FNA.

Test		Sensitivity (95% CI)	Specificity (95% CI)	Positive Predictive Value (95% CI)	Negative Predictive Value (95% CI)	Diagnosis Accuracy (95% CI)
Conventional tools						
CA19-9		37 (19–59)	87 (72–95)	63 (36–85)	69 (54–82)	68 (61–74)
EUS FNA (*n* = 18; PDAC *n* = 15; IPMN *n* = 3)		60 (36–81)	100 (31–99)	100 (60–99)	33 (13–65)	66 (59–73)
Single biomarker based diagnosis method in liquid biopsy						
CTCs	CellSearch^®^ peripheral and/or portal vein (*n* = 30)	32 (15–49)	100	100	35 (18–52)	50 (32–68)
	RosetteSep^TM^ portal vein (*n* = 30)	46 (28–64)	75 (59–90)	84 (71–97)	34 (17–51)	54 (36–72)
	RosetteSep^TM^ peripheral vein (*n* = 42)	50 (35–65)	90 (81–99)	85 (74–96)	63 (48–78)	70 (56–84)
	RosetteSep^TM^ peripheral and/or portal vein (*n* = 52)	59 (46–72)	87 (78–96)	77 (66–88)	75 (63–87)	75 (63–87)
EVs	EVs GPC1 portal vein	46 (27–66)	88 (53–99)	91 (63–99)	36 (20–59)	57 (50–64)
	EVs GPC1 peripheral vein	50 (31–70)	90 (77–99)	79 (58–98)	70 (54–82)	72 (65–78)
	EVs GPC1 peripheral and/or portal vein	64 (43–81)	90 (73–97)	83 (59–94)	76 (59–88)	78 (72–83)
Combined diagnosis methods						
CA19-9 and EUS-FNA		50 (31–70)	92 (78–99)	86 (58–98)	70 (55–83)	74 (67–80)
* CTC RosetteSep^TM^ + EVs GPC1		96 (90–100)	70 (55–83)	70 (55–83)	96 (90–100)	81 (70–93)
* CTC RosetteSep^TM^ + CA19-9 + * EVs GPC1		96 (90–100)	68 (54–83)	68 (54–83)	96 (90–100)	79 (67–92)
* CTC RosetteSep^TM^ + * EVs GPC1 + EUS FNA		96 (90–100)	70 (55–83)	70 (55–83)	96 (90–100)	81 (70–93)
* CTC RosetteSep^TM^ + CA19-9 + * EVs GPC1 + EUS FNA		96 (90–100)	68 (54–83)	68 (54–83)	96 (90–100)	79 (67–92)
* CTC CellSearch^®^ + * CTC RosetteSep^TM^ + * EVs GPC1		100	80 (68–93)	85 (75–96)	100	91 (83–99)

Abbreviations: CI, Confidence interval; CTC, circulating tumor cell; EVs, extracellular vescicles; PDAC, pancreatic ductal adenocarcinoma; IPMN, intraductal papillary and mucinous neoplasm; EUS-FNA endoscopic ultrasound-guided fine needle aspiration. * EVs GPC1, * RosetteSep, and * EVs GPC1, quantification in peripheral and portal vein.

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
