# Peer review of "High Clinical Value of Liquid Biopsy to Detect Circulating Tumor Cells and Tumor Exosomes in Pancreatic Ductal Adenocarcinoma Patients Eligible for Up-Front Surgery"

_cancers, 2019, doi:10.3390/cancers11111656_

Round 1
Reviewer 1 Report
I recently received your paper titled "High-clinical value of liquid biopsy detecting circulating tumor cells and tumor exosomes in pancreatic ductal adenocarcinoma patients eligible for up-front surgery".
Your study focuses on a novel series of biomarkers (circulating tumor exosomes and CTCs) in a setting (pancreatic ductal adenocarcinoma patients) where several diagnostic issues are present owing to the anatomic site of involvement and difficulties in obtaining cytological/histological samples. I agree with you on the fact that this kind of assessment will become widely used in the future and because of that there is a need of papers that focus on this topic.
The methods are explained in detail and the aim of the study is well-explained. The results are also presented in detail and the discussion section cites a few papers focused on the same topic.
I think that the article is acceptable for publication in the present form but, if the authors agree, I would like a few minor additions:
The tests that have been presented seem to have a high value as diagnostic tools (with as much as 100% sensitivity when all 3 methods are used together). I would be interested to know whether they have also some usefulness as prognostic tools: indeed, the authors stated that none of the tests used had a statistically significant impact on prognosis, albeit high values of circulating exosomes and cell-clusters in peripheral blood had lower progression free survival. Do the authors know whether patients submitted to radical surgery had a change of the values of circulating exosomes and cell-clusters after surgery as to make comparison? In the supplementary materials a few clinical characteristics of the patients have been presented. I would like to know whether the PDAC samples that were found at the histologic report were all not-otherwise specified pancreatic ductal adenocarcinoma samples or whether some rarer histology was found (p.e squamous) and if it was associated with a change in the reliability of the tests presented. I would also like to know the site of pancreatic involvement (head-body-tail) of PDAC cancer samples that were resected.
Author Response
We would like to thank the referee for the insightful suggestions that have guided us to improve our manuscript. We have revised the manuscript according to the comments and you will find below a point-by-point description of the modifications.
Reviewer 1:
I recently received your paper titled "High-clinical value of liquid biopsy detecting circulating tumor cells and tumor exosomes in pancreatic ductal adenocarcinoma patients eligible for up-front surgery".
Your study focuses on a novel series of biomarkers (circulating tumor exosomes and CTCs) in a setting (pancreatic ductal adenocarcinoma patients) where several diagnostic issues are present owing to the anatomic site of involvement and difficulties in obtaining cytological/histological samples. I agree with you on the fact that this kind of assessment will become widely used in the future and because of that there is a need of papers that focus on this topic.
The methods are explained in detail and the aim of the study is well-explained. The results are also presented in detail and the discussion section cites a few papers focused on the same topic.
I think that the article is acceptable for publication in the present form but, if the authors agree, I would like a few minor additions:
The tests that have been presented seem to have a high value as diagnostic tools (with as much as 100% sensitivity when all 3 methods are used together).
I would be interested to know whether they have also some usefulness as prognostic tools: indeed, the authors stated that none of the tests used had a statistically significant impact on prognosis, albeit high values of circulating exosomes and cell-clusters in peripheral blood had lower progression free survival.
Do the authors know whether patients submitted to radical surgery had a change of the values of circulating exosomes and cell-clusters after surgery as to make comparison?
RE: We did not take any samples during the follow-up because it was a diagnostic study. However, it is a very good remark that is the subject of a new project in progress.
In the supplementary materials a few clinical characteristics of the patients have been presented. I would like to know whether the PDAC samples that were found at the histologic report were all not-otherwise specified pancreatic ductal adenocarcinoma samples or whether some rarer histology was found (p.e squamous) and if it was associated with a change in the reliability of the tests presented.
RE: It would be very interesting to assess the circulating tumour element presence in the blood of patients bearing rare tumours; however, the histological analysis did not find any rare tumours.
I would also like to know the site of pancreatic involvement (head-body-tail) of PDAC cancer samples that were resected.
RE: This data is available in the table 1, there were 20 tumors located in the head of the pancreas and 2 in the tail of the pancreas. It is now specified in the legend of Table 1 as follows: “Note that Whipple surgery was performed for patients bearing tumours in the head of the pancreas, while left pancreatectomies were performed in patient with tumours in the tail of the pancreas”.
Reviewer 2 Report
The investigators describe a multiplexed, liquid biopsy platform for the diagnosis of early, resectable pancreatic cancer. The platform combines CTC and exosome quantification and detection. A pilot study was performed utilizing blood samples from 22 PDAC patients, 8 IPMN patients and 20 other controls with benign pancreaticobiliary disease or other benign disease.
Comments:
Portal vein blood draws are not likely to prove a useful early detection approach. While an interesting approach for studying biology, this is not a practical platform for a screening tool.
This pilot study consists of very small sample sizes for each cohort. Any biomarker panel optimized in a pilot set needs to be validated in an independent cohort of patients before any determination of performance characteristics can be made. This validation study needs to be performed before publication can be considered.
KRAS mutant CTCs were identified in 4 of 8 IPMN patients, the investigators should confirm presence or absence of KRAS mutation in the resected tissue sample.
While combining CTC and exosome into a single platform is interesting, neither of these approaches is particularly innovative, both have been described in the literature extensively.
Author Response
We would like to thank the referee for the insightful suggestions that have guided us to improve our manuscript. We have revised the manuscript according to the comments and you will find below a point-by-point description of the modifications.
Reviewer 2:
The investigators describe a multiplexed, liquid biopsy platform for the diagnosis of early, resectable pancreatic cancer. The platform combines CTC and exosome quantification and detection. A pilot study was performed utilizing blood samples from 22 PDAC patients, 8 IPMN patients and 20 other controls with benign pancreaticobiliary disease or other benign disease.
Comments:
Portal vein blood draws are not likely to prove a useful early detection approach. While an interesting approach for studying biology, this is not a practical platform for a screening tool.
RE: The feasibility of portal puncture during EUS FNA has been demonstrated in the literature (Catennaci et al 2015) and the approach of the portal vein for blood puncture is achievable by per-cutaneous ultrasound puncture (i.e. portal embolization). This would be applicable to diagnosed PDAC patients, because they would benefit better disease characterization. We do agree that PDAC is not eligible to global screening with the current tools, including portal puncture, as we stated in the discussion (“due to the low lifetime risk of pancreatic cancer (around 1%), population-based screening of unselected individuals is not recommended for this tumor”).
This pilot study consists of very small sample sizes for each cohort. Any biomarker panel optimized in a pilot set needs to be validated in an independent cohort of patients before any determination of performance characteristics can be made. This validation study needs to be performed before publication can be considered.
RE: Our study was a pilot study, as stated in the discussion: “we consider this study as a pilot study, worthy of further validation in bigger cohorts”.
KRAS mutant CTCs were identified in 4 of 8 IPMN patients, the investigators should confirm presence or absence of KRAS mutation in the resected tissue sample.
RE: We agree with the reviewer that this is a very important point. As expected all the IPMNs were KRAS mutant with various MAFs. We now specify this point in the revised manuscript. We provide below the graph gathering the measured MAFs by ddPCR in the IMPNs.
Figure: KRAS mutant detection in the resected IPMN
Figure legend: The red line characterizes the mutant allele frequency positivity threshold; NTC: no template, ADN KRAS m: mutant control DNA, ADN KRAS w: WT DNA, and NN-XXXX XX: DNA from patients with IPMNs.
While combining CTC and exosome into a single platform is interesting, neither of these approaches is particularly innovative, both have been described in the literature extensively.
RE: We agree that taken individually, performing liquid biopsy analysis for PDAC patients is not novel. However, the innovative nature of our work comes from combining detection of circulating biomarkers, which is an emerging strategy to capture rare circulating events with very recent publications.

Reviewer 3 Report
Buscail et al. evaluated combined CTC and exosome detection to diagnose resectable pancreatic cancers.
It is very important to identify new sensitive combination of biomarkers that can be applied to detect PDAC. On this point, this paper is very interesting and attractive. However, some points need to be considered for sophisticated.
All Tables and Figures need to be reconsidered. All was not mature and not enough to explain.
1) Figure 1 is very confusing. Please keep it simple and easy to understand.
2) Please unify the format and expression. It is very difficult to understand because there are letters in the items that contain numbers. For example, Tumor stages, nodes status, glandular differentiation in Table 1.
3) What does author mention and emphasize in Table 2? What does author compare? Could author show ROC in each method?
4) What mean the blue color difference in adenocarcinoma stage in figure 3A?
5) Figure 3B legend says that the bottom ladder indicates dysplasia ranking from 0 for low grade dysplasia to 1 for high grade dysplasia. However, Figure 3B don’t have 0 or 1.
6) Figure 3C show that 4/20 (25%) is positive for RosetteSep and GPC+Exosomes in control groups. This seems to be high false positive rate. What do author think about this result.
7) Author has to mention the color in figure 4C legend.
8) What do X-axis and Y-axis indicate in figure 2A and figure 2B?
Author Response
We would like to thank the referee for the insightful suggestions that have guided us to improve our manuscript. We have revised the manuscript according to the comments and you will find below a point-by-point description of the modifications.
Reviewer 3:
Buscail et al. evaluated combined CTC and exosome detection to diagnose resectable pancreatic cancers.
It is very important to identify new sensitive combination of biomarkers that can be applied to detect PDAC. On this point, this paper is very interesting and attractive. However, some points need to be considered for sophisticated.
All Tables and Figures need to be reconsidered. All was not mature and not enough to explain.
1) Figure 1 is very confusing. Please keep it simple and easy to understand.
RE: We agree and provide a new figure 1 clearly showing the groups compared in this study
2) Please unify the format and expression. It is very difficult to understand because there are letters in the items that contain numbers. For example, Tumor stages, nodes status, glandular differentiation in Table 1.
RE: We agree and have changed the presentation of the data in the table 1 (in red).
3) What does author mention and emphasize in Table 2? What does author compare? Could author show ROC in each method?
RE: We studied the sensitivity and specificity of each parameter and the combination of all available parameters. It is important to show what is currently available to be able to measure the benefit of the liquid biopsy.
The ROC curves of exosomes and CA 19-9 have already been published (Buscail, E. et al.; Transl. Oncol. 2019, 12, 1395–1403). It was possible to obtain an additional ROC curve only for CTC detection by RosetteSepTM method (below), because the other variables are binary and not continuous quantitative. Due to limited editorial space, we did not add this curve in the revised version of the manuscript.
4) What mean the blue color difference in adenocarcinoma stage in figure 3A?
RE: We have clarified the legend as follows:
“In PDAC heat map, the bottom ladder indicates adenocarcinoma stages ranking from 1 to 3 according the stage of the disease (i.e stage 1 light blue; stage 2 blue; stage 3 dark blue)”.
5) Figure 3B legend says that the bottom ladder indicates dysplasia ranking from 0 for low grade dysplasia to 1 for high grade dysplasia. However, Figure 3B don’t have 0 or 1.
RE: We have clarified the legend as follows:
“In IPMN heat map, the bottom ladder indicates dysplasia ranking from 0 (white box) for low grade dysplasia to 1 for high grade dysplasia (blue box)”.
6) Figure 3C show that 4/20 (25%) is positive for RosetteSep and GPC+Exosomes in control groups. This seems to be high false positive rate. What do author think about this result.
RE: We discussed this point in the discussion section as follows: “Previous studies using the ddPCR for identification of KRAS mutant alleles reported false-positive rates in exoDNA varying from 7.4% (4/54) to 20.7% and 25% (17/82 and 3/12)[40,41]. This might be partly explained by the fact that spontaneous somatic mutations are believed to occur in the normal population and healthy tissues [42](39) and the high sensitivity of the PCR-based methods”.
7) Author has to mention the color in figure 4C legend.
RE: We agree and have clarified the legend as follows:
Circulating tumor cell clusters captured from a portal vein sample using the CellSearch system. (CK, cytokeratin; PE, phycoerythrin; DAPI, 4′,6-diamidino-2-phenylindole; DAPI stain is purple and CK stain is green).
8) What do X-axis and Y-axis indicate in figure 2A and figure 2B?
RE: We agree with the reviewer that this information was missing. The legend has been modified as follows: “(X-axis of the left panels corresponding to the HEX, hexachlorofluorescein succinimidyl ester fluorophore) and mutant alleles (Y-axis of the left panels corresponding to the FAM 6-carboxyfluoresceine fluorophore). On the right panels, MAFs are shown for individual results, with the max and the min values of triplicates”.

Round 2
Reviewer 2 Report
The major weakness of this study is that it presents only results from the set of patients used to develop the assay. Without any sort of validation in a separate set of samples, there is no way to assess the validity of this assay. The failure rate of biomarkers without any validation is extraordinarily high. It behoves the investigators to do at least a small amount of validation before claiming an assay is promising.
Author Response
We understand the concern of the reviewer. Our study design did not include the analysis of a validation cohort for time and budget reasons. However, as we fully agree with the reviewer, we have added the following sentence in the conclusion section: "This study is exploratory and in need of further validation on a new cohort of patients with resectable tumors." We also added the following sentence in the abstract: "This exploratory study deserves further validation."
We hope that the reviewer will understand that it would be very long to start a large prospective validation study.
Reviewer 3 Report
The contents are improved.
Author Response
We thank the reviewer for the positive evaluation of our revised version of the manuscript.